# Online psychoeducation and digital assessments as a first step of treatment for borderline personality disorder: A protocol for a pilot randomized controlled trial

**Lois W. Choi-Kain**[1,2☯]*, **Grace E. Murray**[3☯], **Julia Jurist**[1], **Boyu Ren**[1,2], **Laura Germine**[1,2]

**1** McLean Hospital, Belmont, Massachusetts, United States of America, **2** Harvard Medical School, Boston, Massachusetts, United States of America, **3** Boston University, Boston, Massachusetts, United States of America

☯ These authors contributed equally to this work.
* lchoikain@mgb.org

**Data Availability Statement:** No datasets were generated or analysed during the current study. All

## Abstract

### Background

Treatment trials for borderline personality disorder (BPD) have consistently demonstrated that approaches that are diagnostically tailored are superior to those which are not. Currently, gold standard treatments for BPD are highly intensive, lengthy, and specialized, leading to a critical gap between the supply and demand of effective, evidence-based treatment for patients who receive a diagnosis of BPD. Psychoeducation, which is a common component of most treatments known to be effective, is a low-cost, low-burden intervention proven to relieve symptoms. The present study builds on psychoeducation research, assessing online video prescriptions as a means of disseminating information patients need to know about their diagnosis and care.

### Methods

This article presents the study protocol for a safety, feasibility, and preliminary efficacy trial of psychoeducational video prescriptions and online assessment with feedback for newly diagnosed individuals with BPD. We aim to recruit 100 adults recently diagnosed with BPD to be randomly assigned to receive videos about BPD or videos about non-BPD mental health topics that are matched in length in the first step of the study. All participants will complete daily surveys about their emotions, interpersonal interactions, and behaviors, as well as self-report assessments and cognitive tests at 4 different time points. Half of the participants in the intervention group will receive feedback on their symptom ratings and cognitive test performance to assess whether there is incremental value in tailoring this online set of interventions with individualized feedback unique to each participant. This study aims to assess the effects of BPD-focused psychoeducational videos with and without personalized feedback, on BPD and depressive symptom severity as well as core mechanisms of the

relevant data from this study will be made available on request from the author upon study completion.

**Funding:** This study was funded by the Brain and Behavior Research Foundation 2021 NARSAD Young Investigator Grant (30053; https://www.bbrfoundation.org/grants-prizes/bbrf-young-investigator-grants) and the McLean Eric Dorris Translational Research Grant. Funding from both grants was awarded to LW. The funders did not and will not have a role in study design, data collection and analysis, decision to publish, or preparation of the manuscript.

**Competing interests:** The authors have declared that no competing interests exist.

disorder such as loneliness, rejection sensitivity, cognitive control difficulties, and self-clarity. Results will inform efforts to progress to a larger, more definitive trial.

## Trial registration

**Clinical trials registration:** The protocol is registered with ClinicalTrials.gov NCT05358925.

## Introduction

Once thought to be untreatable, borderline personality disorder (BPD) has been legitimized as a valid psychiatric diagnosis with both biological and environmental risk factors, unique longitudinal course marked by high rates of remission and lower rates of recovery, and responsiveness to multiple evidence-based treatments (EBTs) specifically tailored to its core features [1]. Most of these EBTs are intensive, long-term and require extensive training for clinicians. However, meta-analytic studies indicate that duration and intensity do not determine outcome [2]. The specialization and demands of these treatments on both clinicians and patients limit their scalability and contribute to a grave access to care problem [3]. While investigators are testing briefer variants of these treatments that may broaden access over time [4, 5], more resources for clinicians and patients are urgently needed to initiate proper care plans to stabilize patients in acute crises and initiate diagnostically guided care plans efficiently.

Multiple high quality meta-analytic studies have evaluated the efficacy of treatments for BPD that have been subjected to randomized controlled trials [2, 6–8]. The most consistent finding is that psychotherapies designed expressly to treat the problems of BPD result in greater improvements than those that are not diagnostically specific, such as treatment as usual (TAU) and cognitive behavioral therapy (CBT). A common feature of effective treatments is psychoeducation about the disorder, its symptoms, and attention to a core mechanism the treatment proposes to address. This psychoeducational process not only provides much needed information for patients to understand their illness and the problems for which their treatments are prescribed, but also structures the therapeutic alliance, roles, and tasks moving forward. For a disorder distinguished by its unstable relationship vulnerabilities, this structure stabilizes a working framework in the context of expectable interpersonal and behavioral challenges.

Psychoeducation has also been proven as an effective stand-alone intervention. Zanarini and Frankenburg published the results of a trial of psychoeducation involving a sample of 50 women with BPD, randomized to either attend a one day, in-person psychoeducational workshop about BPD or a waitlist control condition. Participants in the workshop condition reported significantly greater reductions in impulsivity and storminess of close relationships than participants in the waitlist condition [9]. Zanarini et al. later assessed the efficacy of a self-guided online psychoeducational program in a randomized control trial of 80 adult women with BPD and found that women in the treatment condition had a significantly greater reduction in all five sectors of borderline symptomatology studied (affective, cognitive, impulsivity, interpersonal, and over-all) [10]. In a study conducted in a community-based outpatient setting, a psychoeducational group conducted over six sessions was tested against a waitlist control condition. Participants of the psychoeducation group had significantly greater reductions in affective, interpersonal, and cognitive symptoms, with almost half the group achieving a greater than 50% reduction in BPD symptoms compared to 3% of the waitlist [11].

Psychoeducational interventions provide a cost-effective first step of care [9]. Training professionals or even non-professional staff to administer interventions of any kind requires resources and skilled human resources. Video prescriptions, which are used across psychiatry and medicine in the routine care of patients [12–15], may eliminate this burdensome requirement for training and clinical face time by enabling clinicians to provide links that patients can access on their own time, watch in the comfort of their chosen surrounding, and review if needed at their leisure. This study aims to extend the existing research on psychoeducation by testing the safety, feasibility, and preliminary efficacy of a suite of short online psychoeducational videos developed by a clinical expert and delivered in a way that directly converses with patients. The control condition videos, produced at the same institution, are matched in length and frequency to the BPD videos and cover mental health topics not specific to BPD. To assess treatment effects and mechanisms, this protocol also includes a comprehensive battery of online assessments, including neuropsychological testing, ecological momentary assessment (EMA), and traditional self-report measures. While the online digital assessments do not utilize gold standard interview-based measures diagnostic of BPD, they employ a valid diagnostic self-report instrument. The absence of a skilled assessor replicates the real-world circumstances of most clinicians who may utilize this digital resource to add an outcomes-based evaluation to the general psychiatric management they provide in their settings.

Online assessments may also serve as an auxiliary intervention in and of itself. Both the regular and daily assessments promote improved self-awareness, which has been shown to be impaired in adults with BPD [16]. While there has been limited research regarding feedback on neuropsychological assessments, numerous published case studies have reported the benefits of feedback, which also increases self-awareness and allows support to be focused on areas of more severe impairment, while enhancing areas of higher ability [17–21]. Feedback on symptom dynamics in a case study increased insight and openness to treatment [22]. Feedback on progress in treatment has been examined as well with promising results for reducing depressive symptoms and improving outcomes [23, 24]. In addition, one non-randomized trial found that receiving feedback on neuropsychological assessment improved patient outcomes on metrics including quality of life and personal mastery [25]. By providing individualized feedback on baseline and longitudinal trends or changes in symptoms and neurocognitive performance, standardized interventions can be tailored so that the patient and clinicians treating them make decisions based on personalized unique data.

## Aims and hypotheses

The specific aims of this study are threefold. First, we aim to test the safety and feasibility of prescribing ten online educational videos about BPD as a first step of care, scalable intervention after diagnostic disclosure. Second, we aim to test the feasibility of an online assessment protocol that incorporates standard self-assessment and neuropsychological testing to assess BPD symptom severity and change, in addition to underlying traits such as rejection sensitivity, and proposed mechanisms of change in treatment such as cognitive control. We will examine variability in social experiences and relative oscillations of mood, cognition, and behavior from the EMA. And lastly, we aim to assess the feasibility and incremental benefit of adding a simple descriptive feedback report as an intervention to personalize the intervention to the individual characteristic of the patient, which might further enhance stabilization of BPD symptoms, depression, anxiety, loneliness, and cognitive dyscontrol. Feasibility will be determined by the completion of recruitment of the sample, rate of completion of viewing prescription videos as well as online assessments, and adequacy of effect sizes for funding application to a larger more definitive trial. Safety will be determined by adverse event monitoring.

Acceptability will be measured with the Client Satisfaction Questionnaire (CSQ-8) by evaluating the group mean in relation to comparable literature. We expect that participants who receive the suite of educational videos about BPD will show greater reductions in borderline and depressive symptom severity post-treatment compared to those who receive a suite of similar control videos of general mental health information. We anticipate that this effect will be mediated by increased knowledge about BPD. We also expect that increased knowledge about BPD, self-clarity, and cognitive control in combination with reduced loneliness will correlate with symptom change. We predict that participants who receive feedback will have reduced symptom severity post-treatment compared to those who do not receive feedback, and that this effect will be mediated by increases in self-clarity and social wisdom (i.e., learning) and decreases in social disconnection. Lastly, we expect that the educational videos and feedback on self-reported symptoms and cognitive performance will yield separate, measurable effects on daily fluctuations in emotional and interpersonal sensitivity in BPD.

## Materials and methods

### Design

This study will follow a sequential multiple randomization trial design with two points of randomization: 1) BPD psychoeducational vs. control videos, and 2) feedback vs. no feedback (Fig 1). This yields three participant groups: (1) *BPD-related videos x neuropsychological and reported symptom feedback*, (2) *BPD-related videos x no feedback*, (3) *control videos x no feedback*. At the first randomization point, randomization will be weighted such that two-thirds of participants enter the BPD-related video arm, and one-third enter the control video arm. At the second randomization point, the participants will be split evenly between the feedback vs. no-feedback conditions. At both randomization points, we will use computer generated block randomization to balance BPD symptom severity and gender. After randomization, participant will be allocated by IDs to the appropriate group by the study research assistant(s). Participants and the statistician will be blind to the assignments.

### Participants

We aim to recruit 100 participants who were recently diagnosed with BPD through advertisements posted both online and in non-psychiatric medical settings, as well as from inpatient psychiatric units. The online advertisements will be posted to the Mass General Brigham participant recruitment platform (i.e., MGB Rally). Study flyers will additionally be posted in hospitals in the greater metropolitan area of Boston and Worcester, Massachusetts. All advertisements will include a brief description of the study, a link or QR code to the virtual eligibility screening survey, and the contact information of the study team. By advertising on the internet and in general healthcare settings to recently diagnosed individuals, we hope to mitigate the bias created by recruiting from subject pools who are already receiving specialized treatment for BPD [3].

To be eligible to participate in this study, participants need to meet all of the following inclusion requirements: (a) reliable access to a smartphone with a data plan for the duration of the study, (b) ability to understand English, (c) age 18 or older, (d) diagnosis of BPD within the past 3 months, (e) ability to complete EMA surveys between 9:00am and 9:00pm on most days, and (f) residence within Massachusetts. Participants meeting either of the following two exclusion criteria will be excluded: (a) having a cognitive disability that impedes ability to participate in the study, or (b) currently experiencing psychiatric symptoms that interfere with the individual's ability to provide consent or complete the research procedures (e.g., acute mania, acute psychosis, or eating disorders threatening medical stability). Participants with

| | STUDY PERIOD | | | | |
| --- | --- | --- | --- | --- | --- |
| | Enrolment | Allocation | Post-allocation | | Close-out |
| **TIMEPOINT\*\*** | $-t_1$ | **0** | $t_1$ | $t_2$ | $t_x$ |
| **ENROLMENT:** | X | | | | |
| **Eligibility screen** | X | | | | |
| **Informed consent** | X | | | | |
| **Allocation** | | X | | | |
| **INTERVENTIONS:** | | | | | |
| *BPD psychoeducational videos* | | ←————————————→ | | | |
| *Non-BPD psychoeducational videos* | | ←————————————→ | | | |
| **ASSESSMENTS:** | | | | | |
| *Demographics* | X | | | | |
| *CPT* | X | | | X | X |
| *RMET* | X | | | | X |
| *BEST* | X | | | | X |
| *DST* | X | | X | X | X |
| *DSMT* | X | | | X | X |
| *Treatment History* | X | | | | X |
| *BPD Knowledge Test* | X | | X | | X |
| *Hospitalization Check-In* | | | X | X | |
| *BSL-23* | X | | X | X | X |
| *PHQ-9* | X | | X | X | X |
| *MARS* | X | | | | X |
| *LS-3* | X | | X | X | X |
| *SD-WISE* | X | | | | X |
| *LPFS* | X | | X | X | X |
| *CSQ-8* | | | X | | |
| *EMA measure* | | ←————————————→ | | | |

**Fig 1. Schedule of enrolment, interventions, and assessments.** -t1 = Timepoint A (baseline); t1 = Timepoint B; t2 = Timepoint C; tx = Timepoint D.

extensive, comprehensive past or current experience with evidence-based psychotherapies for BPD such as dialectical behavior therapy (DBT), mentalization-based treatment (MBT), or transference-focused psychotherapy (TFP) will not be eligible as the study aims to act as a first

step of care intervention for BPD. Eligible participants may continue any other treatments they are receiving for psychiatric conditions throughout the study, which are tracked at baseline and at the last timepoint with a treatment history questionnaire.

Eligible participants will complete a virtual or in person consent discussion with study staff. The research assistant(s) will fully explain the purpose of the research, the study procedures, risks, discomforts, possible benefits, and alternative existing treatments. The study staff will answer any questions from the participant. Participants will be asked to return the signed consent form within 48 hours if they would like to participate, and the form will be signed by the research assistant who completed the consent discussion.

## Interventions

**Psychoeducation.** Two steps of interventions will be employed in this study: psychoeducational videos and feedback on self-reported symptom and trait features, daily EMA, and cognitive task performance. The psychoeducational videos have been filmed for the purpose of this study and feature the principal investigator, an expert in BPD treatment. The videos were shown to three people with lived experience and a family advocate for their perspectives in the development phase. Each of the ten videos is approximately 4–10 minutes long, and covers the following topics: (1) symptoms of BPD, (2) the interpersonal hypersensitivity model of BPD, (3) basic facts about BPD, (4) the long term course of BPD, (5) co-occurring disorders, (6) key principles for recovery, (7) common factors across treatments for BPD, (8) pharmacotherapy for BPD, (9) available psychotherapeutic treatment options for BPD, (10) review of top 10 tips to guide recovery. One video will be delivered to participants in the BPD-related video condition every business day (Mon-Fri) over the course of two weeks.

The videos that will be sent to the participants in the control video condition are matched in length and frequency to the BPD-related videos. The control videos cover aspects of mental and physical health that are not related to BPD, such as diet and exercise, self-compassion, depression, and healthy sleep habits. The suite of control videos was curated from the McLean webinar series, publicly available on the McLean Hospital website for consumers. Links to the control videos can be found in S2 File.

Participants who wish to withdraw their participation in the study may do so at any time and should inform study staff. They will be compensated for the portion of the study they complete.

**Neuropsychological and reported symptom feedback.** Participants across all three arms of the study will complete full-length cognitive tests at four different timepoints: baseline (day 1), time 2 (day 15), time 3 (day 30), and follow-up (day 60). The full-length cognitive tests include: the Gradual Onset Continuous Performance Test (GradCPT) [26], the multiracial version of the Reading the Mind in the Eyes Task (RMET) [26], the Belmont Emotion Sensitivity Test (BEST) [26], the Forward and Backward Digit Span Tests (DST) [27], and the Digit Symbol Matching Test (DSMT) [26]. Additionally, participants will complete 1-minute versions of the CPT and the DSMT daily for 30 days. See the Measures section for more information on each test. Participants who are randomized into the feedback condition will receive an email in the time period between time 2 and time 3 of the study including the results from their cognitive tests and their self-reported BPD symptoms (Fig 2).

## Measures

**Self-reports.** Table 1 outlines the protocol's schedule of assessments. The primary outcome is BPD symptom severity, which will be measured with the Borderline Symptom List 23 (BSL-23) [28]. The BSL-23 is a brief version of the original BSL, which consisted of 95

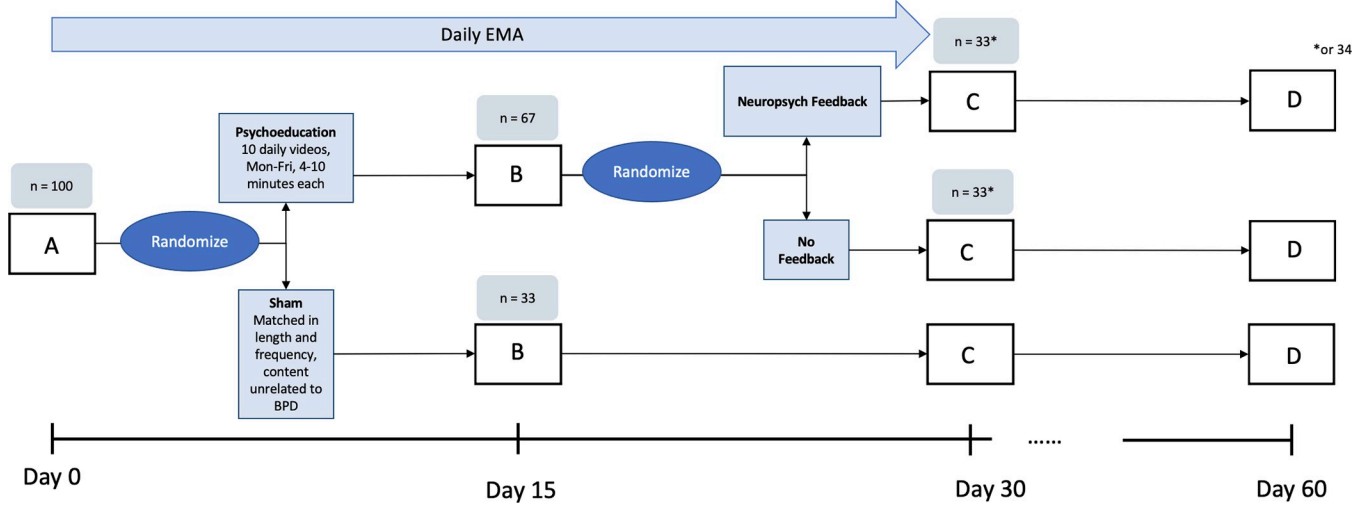

**Fig 2. Study schema.** In the Study Schema, "A" corresponds to Baseline, "B" to Time 2, "C" to Time 3, and "D" to Follow-up.

descriptions of symptoms common in individuals with BPD [28]. The measure has high internal consistency, is strongly correlated with the original BSL, consistently differentiates between individuals with and without BPD, and is sensitive to symptomatic change [28].

The secondary outcome is depressive symptom severity, which will be measured with the Patient Health Questionnaire (PHQ-9) [29]. The PHQ-9 assesses each of the 9 DSM-5 criteria for a major depressive disorder (MD) on a Likert scale from 0 = not at all to 3 = nearly every day. The measure has high internal consistency, differentiates consistently between individuals with and without MDD, has high test-retest reliability, and its criterion validity was established in conjunction with interviews conducted my mental health professionals [29].

Additional self-report measures that will be included in this study are the Three-Item Loneliness Scale (LS-3) [30], the San Diego Wisdom Scale (SD-WISE) [31], and the Level of Personality Functioning Scale-Brief (LPFS-B) [32]. The LPFS assesses criterion A's domains of self and interpersonal functioning, comprising a measure of general personality pathology and impairment [29]. The three aforementioned scales are all psychometrically validated with acceptable internal reliability and construct validity [30–32]. The Client Satisfaction

**Table 1. Schedule of assessments.**

| Timepoint A (Baseline) | Timepoint B | Timepoint C | Timepoint D |
|---|---|---|---|
| • Demographics<br>• Treatment history<br>• BPD symptom severity (BSL-23)<br>• Depressive symptom severity (PHQ-9)<br>• BPD Knowledge Test<br>• Rejection sensitivity (MARS)<br>• Loneliness (LS-3)<br>• Social wisdom (SD-WISE)<br>• Personality functioning (LPFS)<br>• Cognitive control (CPT)<br>• Social understanding; theory of mind (RMET)<br>• Emotion perception (BEST)<br>• Working memory; attention (DST)<br>• Processing speed; short term memory (DSMT) | • BPD symptom severity (BSL-23)<br>• Depressive symptom severity (PHQ-9)<br>• BPD Knowledge Test Hospitalization check-in<br>• Loneliness (LS-3)<br>• Personality functioning (LPFS)<br>• Working memory; attention (DST)<br>• Acceptability (CSQ-8) | • BPD symptom severity (BSL-23)<br>• Depressive symptom severity (PHQ-9)<br>• Hospitalization check-in<br>• Loneliness (LS-3)<br>• Personality functioning (LPFS)<br>• Cognitive control (CPT)<br>• Working memory; attention (DST)<br>• Processing speed; short term memory (DSMT) | • Treatment history<br>• BPD symptom severity (BSL-23)<br>• Depressive symptom severity (PHQ-9)<br>• BPD Knowledge Test<br>• Rejection sensitivity (MARS)<br>• Loneliness (LS-3)<br>• Social wisdom (SD-WISE)<br>• Personality functioning (LPFS)<br>• Cognitive control (CPT)<br>• Social understanding; theory of mind (RMET)<br>• Emotion perception (BEST)<br>• Working memory; attention (DST)<br>• Processing speed; short term memory (DSMT) |

Questionnaire (CSQ-8) [33] will also be included as an acceptability questionnaire, including questions about satisfaction and whether the intervention has helped them deal more effectively with their problems.

Participants will also report their mental health treatment history on a self-report measure that was adapted from the Background Information Schedule for this study [34], report hospitalizations that have occurred since the last data collection time-point, and complete an assessment of their level of knowledge about BPD and its treatment developed by the study team.

**Ecological momentary assessment.** In the first 30 days of the study, all participants will respond to a daily EMA survey containing questions about the participant's immediate surroundings and functioning, as well as their emotional, interpersonal, and behavioral reactions to recent stressful or supportive social interactions. Each EMA survey will be accompanied by a 1-minute version of the CPT and the DSMT. The survey will be sent to the participants at a randomly generated time between 9:00am and 9:00pm and will take approximately 5–6 minutes.

**Cognitive tests.** In the GradCPT, participants are instructed to press a button when a city scene is displayed on the screen, and to refrain from pressing a button when a mountain scene is displayed. The images fade from one into the other, with no pause between trials. City scenes appear in 80–90% of trials, and mountain scenes in only 10–20% of trials. This task measures sustained attention, response inhibition, and cognitive control [26, 27].

In the DST, participants are asked to recall lists of digits in the same (forward condition) or reverse (backward condition) order that they were originally presented, and an individual's scores on the forward and backward conditions represents the length of the longest list of digits they were able to accurately recall and type at least once. Scores in the forward condition can indicate attention/concentration ability, while scores in the backward condition can indicate working memory [35].

In the BEST, participants choose which of two faces presented on the screen is more emotional for three different affect conditions: anger, fear, and happiness. All three affect conditions assess social communication ability (i.e., ability to receive social communication from facial expressions) and ability to understand mental states. The anger and fear conditions also measure potential threat, while the happiness condition measures positive valence [26].

In the RMET, participants are shown images of only the eye-region of different faces and asked to select one of four emotion words on the screen to describe the face's emotional expression. This task assesses social communication ability (i.e., ability to receive social communication from facial expressions) [26].

In the DSMT, participants are asked to use a symbol-number key shown on screen to match as many symbols and numbers as possible in 90 seconds. This test measures cognitive processing speed and visual short-term memory [26].

## Data management, monitoring, and safety

**Data management.** Data will be securely stored with each participant assigned a participant ID for confidentiality and all personal identifying information filed separately. Physical study data will be locked in separate files. Electronic data will be password protected on encrypted devices, labeled with the participant ID number. All neuropsychological data will be coded with unique numbers and be temporarily stored in the Amazon Relational Database Service, a cloud-based data storage service connected to Amazon Web Services (i.e., TestMy-Brain's secure database) before being downloaded. After download, data will be stored on study staff computers in locked offices. Access to all data will be limited to IRB-approved study staff.

**Data monitoring.** The PIs are responsible for continuous monitoring of data and safety of participants in the study. Adverse events will be documented and reported as required by the MGB Human Research Committees (MGBHRC) policy. The PIs will report to the MGBHRC any unanticipated problems and adverse events that occur during the study, after study completion, or after participant withdrawal or completion. Reports will be submitted within 5 working days/7 calendar days of the date the investigator first becomes aware of the problem. At the time of the continuing review, we will provide the MGB IRB with a summary of any unexpected and related adverse events as well as any other unanticipated problems that occurred since the last continuing review.

**Safety.** Daily assessments include a reminder for patients that study staff will not monitor data immediately and if they need immediate attention or feel unsafe, they should call 911 or go to their local Emergency Department. Participants are told the aforementioned safety information during the consenting process as well.

## Statistical methods

We expect a small to medium effect on BPD symptom severity, in line with effect sizes in the meta-analytic review of BPD's effective interventions [2]. With a significance level of .05 and power of .8, the necessary sample size per condition to detect a small effect (Cohen's $d$ = .2) using two-sample t-test is 394, and to detect a medium effect (Cohen's $d$ = .4), $n$ = 99. As a pilot study, this design is not adequately powered to detect the expected small to medium effect-size; therefore, we will conduct exploratory analyses in preparation for a future, large-scale investigation. Efficacy in this pilot trial will be estimated with 95% confidence intervals. We will 1) test associations between BPD and depressive symptoms and treatments in step 1 (video intervention) and step 2 (neuropsychological testing and symptom rating feedback) separately. We will first examine the normality of the BPD and depressive symptoms to determine the appropriate test–we will perform a two-sample t-test if normality holds and a Wilcoxon rank sum test otherwise; 2) explore replacing binary treatment assignment with the actual number of educational videos one watched in a secondary analysis and test its association with BPD and depressive outcomes in step 1. We will use a general linear model for the analysis. Appropriate transformation of the outcome will be performed to handle potential non-normality; 3) conduct a mediation analysis using the R package *mediation* to estimate the direct effect of each treatment on the primary and secondary outcomes, as well as the indirect effects via the proposed mediators; 4) perform longitudinal data analysis with linear mixed models to examine the population-level effects of different treatment combinations on outcomes of interest, such as cognitive control while accounting for the correlation between repeated measures; and 5) test whether differential responses to step 1 are associated with changes in effectiveness of treatments in step 2 by considering a linear regression model where the change of BPD symptoms from baseline to step 2 is regressed on the change of BPD symptoms from baseline to step 1. We will use multiple imputations through chained equations to handle missing data and will perform sensitivity analysis to examine the robustness of the results when the anchoring missing data mechanism is either missing at random (MAR) or missing not at random (MNAR). For multiple imputations based on MAR assumption, we will also compare the results with those from directly applying linear mixed models on the available observations whenever appropriate. We will focus a flexible MNAR anchoring mechanism, no self-censoring [36] and introduce a unified sensitivity parameter which encodes the dependence of missingness to the underlying observation [37, 38]. We will use R package *mice* to perform imputations [39] and the results from each imputation will be combined using Rubin's rule [40].

### Status

The study was approved by the Mass General Brigham IRB (2022000892) and began recruitment and data collection in June 2022.

## Discussion

This pilot study aims to preliminarily assess safety, feasibility, and preliminary efficacy of online psychoeducation and assessment with individualized feedback which primary care providers and generalist mental health clinicians might be able to prescribe for patients upon diagnostic disclosure before uptake or enrollment in further treatment. Many patients and clinicians are unable to access BPD-specific care due to a lack of availability in their locale, stigma, or prohibitive costs in treatments not automatically covered by insurance [3]. Similarly, many primary care physicians (PCPs), psychopharmacology prescribers, emergency room clinicians, or generalist mental health clinicians are unable to provide referrals to gold standard treatment for their patients with BPD. Instead of relying on a time and resource intensive process of training clinicians and staff to provide psychoeducation and relevant assessments for optimizing care, this set of digital interventions provides an opportunity to systematize a readily available system of meeting needs in the current climate of pervasive shortages of mental health services to meet demands.

Delivering the psychoeducation asynchronously through videos sent directly to participants may have some advantages over both in-person teaching and website-based programs. The asynchronous nature of the intervention avoids two significant sources of variability in treatment response: the therapist themself and the therapist's role in the alliance [41, 42]. This may be particularly helpful to individuals with BPD, who are hypersensitive in interpersonal relationships and interactions [43]. Providing videos directly to patients via a daily email or text message may motivate or remind participants to continue with the intervention, rather than relying on the participant to self-direct to a specific webpage. It also adds a low level of direct communication, which may increase participant buy-in, especially if they may be prone to avoid social contact out of shame, paranoia, mistrust, or logistical barriers such as work, childcare, or travel barriers to treatment access. By creating widely accessible, user-friendly digital resources such as these, and testing their efficacy, we may be able to broadly augment existing care with a specialized BPD framework that can augment usual good clinical care available in most settings.

This study will produce preliminary data to support future, larger scale research on psychoeducational video prescriptions as a scalable intervention to reduce BPD symptomatology. This study will also contribute to existing research on neuropsychological and reported symptom feedback as an intervention.

## Supporting information

**S1 File. Completed SPIRIT checklist.**
(DOC)

**S2 File. Control videos.**
(DOCX)

**S3 File. Detailed protocol.**
(DOCX)

## Acknowledgments

We would like to express our gratitude to our funding sources the Brain & Behavior Research Foundation, Families for BPD Research, and the Eric Dorris Translational Research Award for providing this opportunity to study this approach to broadening access to care. We also thank Kerry Ressler M.D. Ph.D. for his mentorship in our "blue sky" idea for our NARSAD Young Investigator grant which made this effort possible. Lastly, we thank Rebbie Ratner and her team in filming and producing our psychoeducational video series.

## Author Contributions

**Conceptualization:** Lois W. Choi-Kain, Grace E. Murray, Boyu Ren, Laura Germine.

**Funding acquisition:** Lois W. Choi-Kain, Grace E. Murray.

**Methodology:** Lois W. Choi-Kain, Grace E. Murray, Boyu Ren, Laura Germine.

**Writing – original draft:** Lois W. Choi-Kain, Grace E. Murray.

**Writing – review & editing:** Lois W. Choi-Kain, Julia Jurist, Boyu Ren, Laura Germine.

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
