## [Decision Letter · Decision Letter 0]

13 Sep 2023

PONE-D-23-21097Online psychoeducation and digital assessments as a first step of treatment for borderline personality disorder: A protocol for a pilot studyPLOS ONE

Dear Dr. Choi-Kain,

Thank you for submitting your manuscript to PLOS ONE. After careful consideration, we feel that it has merit but does not fully meet PLOS ONE’s publication criteria as it currently stands. Therefore, we invite you to submit a revised version of the manuscript that addresses the points raised during the review process.

We look forward to receiving your revised manuscript.

Kind regards,

Stephan Doering, M.D.

Academic Editor

PLOS ONE

3. We note that the original protocol that you have uploaded as a Supporting Information file contains an institutional logo. As this logo is likely copyrighted, we ask that you please remove it from this file and upload an updated version upon resubmission.

Reviewers' comments:

Reviewer's Responses to Questions

**Comments to the Author**

1. Does the manuscript provide a valid rationale for the proposed study, with clearly identified and justified research questions?

Reviewer #1: Yes

Reviewer #2: Partly

2. Is the protocol technically sound and planned in a manner that will lead to a meaningful outcome and allow testing the stated hypotheses?

Reviewer #1: Yes

Reviewer #2: Partly

3. Is the methodology feasible and described in sufficient detail to allow the work to be replicable?

Reviewer #1: Yes

Reviewer #2: Yes

4. Have the authors described where all data underlying the findings will be made available when the study is complete?

Reviewer #1: Yes

Reviewer #2: No

5. Is the manuscript presented in an intelligible fashion and written in standard English?

Reviewer #1: Yes

Reviewer #2: Yes

6. Review Comments to the Author

You may also provide optional suggestions and comments to authors that they might find helpful in planning their study.

Reviewer #1: How will the feasibility of the study be assessed? Will compliance also be measured?

The study is underpowered, so the author needs to clarify that no hypotheses will be tested. However, efficacy can still be estimated with 95% confidence intervals.

More detailed information is required for the analyses. For instance, in Analysis 1, how will the associations between BPD and treatment be examined? Will the difference in the mean change from baseline be utilized? What specific model will be employed?

Missing at random can be addressed through mixed models. Conversely, for data that is not missing at random, multiple imputation offers a viable solution. Additional specifics are necessary. Furthermore, the sample size should account for potential attrition.

Reviewer #2: This manuscript reports the protocol for a pilot RCT of (i) psychoeducation videos on BPD vs videos on general mental health for 100 people with recently diagnosed BPD; and (ii) assessment and feedback vs no feedback for ~66 people who were randomised to receive the BPD psychoed videos. The study aims to evaluate safety and feasibility, with a preliminary efficacy evaluation (for BPD and depression severity). The development and evaluation of low-cost, scalable BPD interventions is sorely needed. Other study strengths include: examination of mechanism of change; an active comparison/control condition matched to the experimental intervention; and multi modal assessment. I believe it is appropriate to use a well-validated self-report BPD measure for a pilot trial.

However, there are some issues to be addressed before this protocol can be accepted for publication:

Lines 63-66

In the background section you state that intervention approaches that are diagnostically tailored are superior to those that aren’t. This study tests a diagnostic-specific intervention (videos on BPD) vs a generic intervention (videos on general mental health and wellbeing). However the generic (control) intervention includes information about a diagnosis (depression), and depression is a secondary outcome of the trial, thus the nature of the control may confound the interpretation of the results. i.e. if no difference in depression severity between 2 interventions, could be because control included depression content. This needs to be addressed in the paper, particularly given the high rates of concurrent diagnoses of BPD & MDD (~70-80% in tertiary psychiatric settings).

Line 95

The BPD videos were developed by a clinical expert and the control videos were sources form a McLean webinar series. Were the perspectives of those with lived experience of diagnosis and treatment for personality disorder included in the development and/or testing of these resources? If not, could this limitation be stated in the paper, and addressed in a full-scale RCT of the intervention?

Lines 105-115

In this section (and throughout the manuscript) the term neuropsychological feedback is used. It seems that the literature in this section primarily refers to traditional neuropsych outcomes (e.g. memory, attention, etc). Is there evidence for the effect of feedback regarding change in clinical symptoms? This should be included, or its absence noted. It seems that you are providing feedback from self-report measure (BSL) and neuropsych/cognitive measures, so I’d postit that a term other than neuropsychological feedback would enhance the clarity of the manuscript.

Lines 117 – 122

This study tests safety and feasibility of an online BPD videos (aim 1) and feasibility of an assessment protocol (aim 2), and feasibility of a feedback report (aim 3), yet in the manuscript the authors haven’t detailed the criteria/thresholds they are using to judge the safety and feasibility for the interventions. E.g. group mean exceeding a CSQ score of X? < X discontinuations? X hospital admissions deemed possibly related to intervention?

Line 117

The first aim states that you’ll be testing BPD videos as a first step of care after diagnostic disclosure. The inclusion/exclusion criteria only refer to recent diagnosis and are silent on whether the participants are in treatment (and what stage of treatment) or not. While I believe it’s appropriate to say that you envision this being used as a first step of treatment (in intro and discussion), this study design doesn’t address this.

Line 140

Please include how you are generating the randomisation sequence (e.g. computer generated by independent statistician) and who performing the randomisation (e.g. randomised by with a digital program; sealed envelopes). Who performs the randomisation?

Line 2010

Intervention discontinuation criteria and/or study withdrawal criteria is missing. E.g. if a participant informs that they no long wish to receive the intervention and/or participate in the research assessments. Are there any others that haven’t been reported?

Line 212 & Line 209

I believe the same subjective, self-report measure (BSL) is used as both as part of the intervention (i.e. the step 2 feedback) and as the outcome measure for the effectiveness of the feedback. Could you include comment on the impact of this?

Line 300-304

Earlier in manuscript you state that you’re also testing the effect of the intervention on depression (secondary outcome) as well as BPD (primary outcome but test 1) and 2) doesn’t mention depression.

300-312

Please include how you are using EMA as an outcome in your analyses in this section (as you’ve mentioned this in your aims & hypotheses section earlier on).

Editing/typos:

Line 75 – attend or attending (not attended).

Line 251 – typo. Should be mountain not mount.

Line 283 – policy not polity.

Optional suggestions and comments:

Line 3

Add that it’s a protocol for a pilot randomised controlled trial in the title.

Lines 73-86

For clarity, it’d be worth mentioning that the two Zanarini studies recruited women from the general community (not health settings). Also, Ridolfi’s study recruited a sample from a health setting (an outpatient psychiatry service), but you state this was conducted in the community.

Lines 194-198

It’d be helpful to include a breakdown of the 10 video topics (similar to the breakdown provided for BPD vids).

Line 208

Is the feedback email content and / or automated? Or will a staff member need to draft a report to email?

7. PLOS authors have the option to publish the peer review history of their article (what does this mean?). If published, this will include your full peer review and any attached files.

Reviewer #1: No

Reviewer #2: No

---

## [Author Response · Author response to Decision Letter 0]

10 Oct 2023

Reviewer #1: 

How will the feasibility of the study be assessed? Will compliance also be measured?

Thank you for pointing this out that we have not clarified this as an assessed outcome.

• We added to manuscript line 130-135: “Feasibility will be determined by the completion of recruitment of the sample, rate of completion of viewing prescription videos as well as online assessments, and adequacy of effect sizes for funding application to a larger more definitive trial. Safety will be determined by adverse event monitoring. Acceptability will be measured with the Client Satisfaction Questionnaire (CSQ-8) by evaluating the group mean in relation to comparable literature.”

The study is underpowered, so the author needs to clarify that no hypotheses will be tested. However, efficacy can still be estimated with 95% confidence intervals.

Thanks for this adjustment for being more precise for the data plan in our pilot trial protocol.

• We added to manuscript line 318-321: “As a pilot study, this design is not adequately powered to detect the expected small to medium effect-size; therefore, we will conduct exploratory analyses in preparation for a future, large-scale investigation. Efficacy in this pilot trial will be estimated with 95% confidence intervals.”

More detailed information is required for the analyses. For instance, in Analysis 1, how will the associations between BPD and treatment be examined? Will the difference in the mean change from baseline be utilized? What specific model will be employed?

We took this opportunity to fill in this detail with the following changes:

• We revised manuscript lines 324-338: 

o 1) “We will first examine the normality of the BPD and depressive symptoms to determine the appropriate test – we will perform a two-sample t-test if normality holds and a Wilcoxon rank sum test otherwise”

o 2) “We will use a general linear model for the analysis. Appropriate transformation of the outcome will be performed to handle potential non-normality”

o 3) “using the R package mediation”

o 4) “perform longitudinal data analysis with linear mixed models to examine the population-level effects of different treatment combinations on outcomes of interest, such as cognitive control while accounting for the correlation between repeated measures”

o 5) “by considering a linear regression model where the change of BPD symptoms from baseline to step 2 is regressed on the change of BPD symptoms from baseline to step 1”

Missing at random can be addressed through mixed models. Conversely, for data that is not missing at random, multiple imputation offers a viable solution. Additional specifics are necessary. Furthermore, the sample size should account for potential attrition. 

Agreed, we’ve made the following additions:

• At lines 338-346: “We will use multiple imputations through chained equations to handle missing data and will perform sensitivity analysis to examine the robustness of the results when the anchoring missing data mechanism is either missing at random (MAR) or missing not at random (MNAR). For multiple imputations based on MAR assumption, we will also compare the results with those from directly applying linear mixed models on the available observations whenever appropriate. We will focus a flexible MNAR anchoring mechanism, no self-censoring [33], and introduce a unified sensitivity parameter which encodes the dependence of missingness to the underlying observation [34, 35]. We will use R package mice to perform imputations [36] and the results from each imputation will be combined using Rubin’s rule [37].”

Reviewer #2: This manuscript reports the protocol for a pilot RCT of (i) psychoeducation videos on BPD vs videos on general mental health for 100 people with recently diagnosed BPD; and (ii) assessment and feedback vs no feedback for ~66 people who were randomised to receive the BPD psychoed videos. The study aims to evaluate safety and feasibility, with a preliminary efficacy evaluation (for BPD and depression severity). The development and evaluation of low-cost, scalable BPD interventions is sorely needed. Other study strengths include: examination of mechanism of change; an active comparison/control condition matched to the experimental intervention; and multi modal assessment. I believe it is appropriate to use a well-validated self-report BPD measure for a pilot trial. However, there are some issues to be addressed before this protocol can be accepted for publication:

Thank you for validating the major purposes of this study and its aims. We also appreciate the specificity of the following comments we address here:

Lines 63-66

In the background section you state that intervention approaches that are diagnostically tailored are superior to those that aren’t. This study tests a diagnostic-specific intervention (videos on BPD) vs a generic intervention (videos on general mental health and wellbeing). However the generic (control) intervention includes information about a diagnosis (depression), and depression is a secondary outcome of the trial, thus the nature of the control may confound the interpretation of the results. i.e. if no difference in depression severity between 2 interventions, could be because control included depression content. This needs to be addressed in the paper, particularly given the high rates of concurrent diagnoses of BPD & MDD (~70-80% in tertiary psychiatric settings).

We appreciate this concern as we have observed most BPD samples contain very few individuals who do not have concurrent or lifetime depression. We could most certainly also assess whether depression as an outcome decreases more in the control arm since there is one video on depression in addition to others on general mental health literacy such as self-care, and coping. We can compare whether or not the control group has a greater decline in PHQ-9 scores compared to the BPD psychoeducational arm, but discerning the effects of decreases in BPD scores on depression versus those related to the video interventions will be difficult in this small a sample.

Line 95

The BPD videos were developed by a clinical expert and the control videos were sources form a McLean webinar series. Were the perspectives of those with lived experience of diagnosis and treatment for personality disorder included in the development and/or testing of these resources? If not, could this limitation be stated in the paper, and addressed in a full-scale RCT of the intervention?

Yes, we did consult people with lived experience. Rebbie Ratner from BorderlinerNotes was our producer of the videos and is a lived experience expert and also trialed the videos with her subjects to test them. We also reviewed them with the leaders of Families for BPD Research Organization.

• We added to manuscript line 201-203: “The videos were shown to three people with lived experience and a family advocate for their perspectives in the development phase.”

Lines 105-115

In this section (and throughout the manuscript) the term neuropsychological feedback is used. It seems that the literature in this section primarily refers to traditional neuropsych outcomes (e.g. memory, attention, etc). Is there evidence for the effect of feedback regarding change in clinical symptoms? This should be included, or its absence noted. It seems that you are providing feedback from self-report measure (BSL) and neuropsych/cognitive measures, so I’d posit that a term other than neuropsychological feedback would enhance the clarity of the manuscript.

We agree and appreciate reviewer 2 catching this discrepancy. The digital neuropsychological platform was created by one of the authors based on traditional measures and test, but more current papers on this novel platform are in development. We’ve responded to these concerns with the following:

• We changed “neuropsychological feedback” more generally to “feedback” or “neuropsychological and reported symptom feedback” throughout the manuscript, except in lines 144-145 where we revised this phrase to“feedback on self-reported symptoms and cognitive performance.”

• We added to manuscript line 110-113: “Feedback on symptom dynamics in a case study increased insight and openness to treatment [22]. Feedback on progress in treatment has been examined as well with promising results for reducing depressive symptoms and improved outcomes [23, 24].”

• Lastly, we added several citations with evidence for feedback and clinical change in symptoms:

o Kroeze R, van der Veen DC, Servaas MN, Bastiaansen JA, Oude Voshaar RC, Borsboom D, Ruhe HG, Schoevers RA, Riese H. Personalized Feedback on Symptom Dynamics of Psychopathology: A Proof-of-Principle Study. J Pers Oriented Res. 2017 Nov 1;3(1):1-10. doi: 10.17505/jpor.2017.01.

o Newnham EA, Hooke GR, Page AC. Progress monitoring and feedback in psychiatric care reduces depressive symptoms. J Affect Disorder. 2010 Dec 1;127(1-3):139-46. https://doi.org/10.1016/j.jad.2010.05.003

o Hawkins EJ, Lambert MJ, Vermeersch DA, Slade KL, Tuttle KC. The therapeutic effects of providing patient progress information to therapists and patients. Psychother Res. 2004;14(3):308-327. https://doi.org/10.1093/ptr/kph027

Lines 117 – 122

This study tests safety and feasibility of an online BPD videos (aim 1) and feasibility of an assessment protocol (aim 2), and feasibility of a feedback report (aim 3), yet in the manuscript the authors haven’t detailed the criteria/thresholds they are using to judge the safety and feasibility for the interventions. E.g. group mean exceeding a CSQ score of X? < X discontinuations? X hospital admissions deemed possibly related to intervention?

We address this concern above in the first response to reviewer 1. In addition:

• We added to manuscript line 130-135: “Feasibility will be determined by the completion of recruitment of the sample, rate of completion of viewing prescription videos as well as online assessments, and adequacy of effect sizes for funding application to a larger more definitive trial. Safety will be determined by adverse event monitoring. Acceptability will be measured with the Client Satisfaction Questionnaire (CSQ-8) by evaluating the group mean in relation to comparable literature.”

Line 117

The first aim states that you’ll be testing BPD videos as a first step of care after diagnostic disclosure. The inclusion/exclusion criteria only refer to recent diagnosis and are silent on whether the participants are in treatment (and what stage of treatment) or not. While I believe it’s appropriate to say that you envision this being used as a first step of treatment (in intro and discussion), this study design doesn’t address this.

We agree and this is a part of our recruitment strategy. In addition we use treatment history and treatment at each interval in the analysis.

• We added to manuscript line 181-187: “Participants with extensive, comprehensive past or current experience with evidence-based psychotherapies for BPD such as dialectical behavior therapy (DBT), mentalization-based treatment (MBT), or transference-focused psychotherapy (TFP) will not be eligible as the study aims to act as a first step intervention for BPD. Eligible participants may continue any other treatments they are receiving for psychiatric conditions throughout the study which are tracked at baseline and at the last timepoint with a treatment history questionnaire.”

Line 140

Please include how you are generating the randomisation sequence (e.g. computer generated by independent statistician) and who performing the randomisation (e.g. randomised by with a digital program; sealed envelopes). Who performs the randomisation?

Yes this would require clarification:

• We added to manuscript line 156-159: “At both randomization points, we will use computer generated block randomization in R, balancing BPD symptom severity and gender. The study research assistant(s) will enter the results into the survey software (redcap) to assign participants by IDs to the appropriate group. Participants and the statistician will be blind to the assignment.”

Line 210

Intervention discontinuation criteria and/or study withdrawal criteria is missing. E.g. if a participant informs that they no long wish to receive the intervention and/or participate in the research assessments. Are there any others that haven’t been reported?

Agree with this gap and have addressed here:

• We added to manuscript line 216-218: “Participants who wish to withdraw their participation in the study may do so at any time and should inform study staff. They will be compensated for the portion of the study they complete.”

Line 212 & Line 209

I believe the same subjective, self-report measure (BSL) is used as both as part of the intervention (i.e. the step 2 feedback) and as the outcome measure for the effectiveness of the feedback. Could you include comment on the impact of this?

This is an interesting problem we did not consider. Repeated measurements do affect responding over each administration, and it is likely assessments themselves increase the participant’s awareness of their problems in a way that likely enhances self-clarity as a pseudo-psychoeducational intervention in and of itself. Since the control group and the BPD psychoeducation group without feedback also gets repeated administration of this measure, we feel the additional layer of feedback may provide a more isolated effect of feedback rather than exposure to the measure. We can look at this in the results reporting but given the pilot nature of this study this item while interesting may be one we delay exploring until we can get funding for a more definitive study. We appreciate reviewer 2 raising our awareness to this.

Line 300-304

Earlier in manuscript you state that you’re also testing the effect of the intervention on depression (secondary outcome) as well as BPD (primary outcome but test 1) and 2) doesn’t mention depression.

Yes this is a secondary issue in our study which we will acknowledge:

• We added to manuscript line 322-328: “We will 1) test association between BPD and depressive symptoms and treatments… 2) explore replacing binary treatment assignment with the actual number of educational videos one watched in a secondary analysis and test its association with BPD and depressive outcomes in step 1…”

Line 300-312

Please include how you are using EMA as an outcome in your analyses in this section (as you’ve mentioned this in your aims & hypotheses section earlier on).

Thanks for helping us complete that loop by bringing that to our attention:

• We added to manuscript line 332-334: “… to examine the population-level effects of different treatment combinations on outcomes of interest, such as cognitive control”

• Earlier in the manuscript, we added line 125-127: “We will examine variability in social experiences and relative oscillations of mood, cognition, and behavior from the EMA.”

Thank you for the following detailed edits which we corrected:

Editing/typos:

Line 75 – attend or attending (not attended).

Line 251 – typo. Should be mountain not mount.

Line 283 – policy not polity.

Optional suggestions and comments:

Line 3

Add that it’s a protocol for a pilot randomised controlled trial in the title.

Thank you we’ve done that.

Lines 73-86

For clarity, it’d be worth mentioning that the two Zanarini studies recruited women from the general community (not health settings). Also, Ridolfi’s study recruited a sample from a health setting (an outpatient psychiatry service), but you state this was conducted in the community.

Agreed:

• We added to manuscript line 82: “a community-based outpatient setting”

Lines 194-198

It’d be helpful to include a breakdown of the 10 video topics (similar to the breakdown provided for BPD vids).

Thank you this is interesting and helpful to readers:

• We made a supplementary file 2 and added to manuscript lines 214-215: “Links to the control videos can be found in Supplement 2.”

Line 208

Is the feedback email content and / or automated? Or will a staff member need to draft a report to email?

The feedback is both automated through data summarizing and figure creation but adjusted individually as a mode for us as the study team to think through how we can do this on a larger scale. The limited funding of this study prevented the employment of a more sophisticated computerized approach which hopefully we can incorporate with a more definitive trial.

---

## [Editor Report · Decision Letter 1]

30 Oct 2023

Online psychoeducation and digital assessments as a first step of treatment for borderline personality disorder: A protocol for a pilot randomized controlled trial

PONE-D-23-21097R1

Dear Dr. Choi-Kain,

We’re pleased to inform you that your manuscript has been judged scientifically suitable for publication and will be formally accepted for publication once it meets all outstanding technical requirements.

Kind regards,

Stephan Doering, M.D.

Academic Editor

PLOS ONE

---

## [Editor Report · Acceptance letter]

22 Nov 2023

PONE-D-23-21097R1 

Online psychoeducation and digital assessments as a first step of treatment for borderline personality disorder: A protocol for a pilot randomized controlled trial 

Dear Dr. Choi-Kain:

I'm pleased to inform you that your manuscript has been deemed suitable for publication in PLOS ONE. Congratulations! Your manuscript is now with our production department. 

Kind regards, 

on behalf of

Professor Stephan Doering 

Academic Editor

PLOS ONE